# Photoreactive Coating Material as an Effective and Durable Antimicrobial Composite in Reducing Bacterial Load on Surfaces in Livestock

**DOI:** 10.3390/biomedicines10092312

**Published:** 2022-09-17

**Authors:** Ádám Kerek, Mátyás Sasvári, Ákos Jerzsele, Zoltán Somogyi, László Janovák, Zsolt Abonyi-Tóth, Imre Dékány

**Affiliations:** 1Department of Pharmacology and Toxicology, University of Veterinary Medicine, István Street 2, H-1078 Budapest, Hungary; 2Department of Physical Chemistry and Materials Science, University of Szeged, Rerrich Béla tér 1, H-6720 Szeged, Hungary; 3Department of Biomathematics and Informatics, University of Veterinary Medicine, István Street 2, H-1078 Budapest, Hungary

**Keywords:** titanium dioxide (TiO_2_), zinc oxide (ZnO), photoreactive coating, *Escherichia coli*

## Abstract

Titanium dioxide (TiO_2_) is a well-known photocatalytic compound that can be used to effectively reduce the presence of pathogens in human and animal hospitals via ROS release. The aim of this study was to investigate the efficacy of a polymer-based composite layer containing TiO_2_ and zinc oxide (ZnO) against *Escherichia coli* (*E. coli*) of animal origin. We showed that the photocatalyst coating caused a significant (*p* < 0.001) reduction in pathogen numbers compared to the control with an average reduction of 94% over 30 min. We used six light sources of different wattages (4 W, 7 W, 9 W, 12 W, 18 W, 36 W) at six distances (35 cm, 100 cm, 150 cm, 200 cm, 250 cm, 300 cm). Samples (*n* = 2160) were taken in the 36 settings and showed no significant difference in efficacy between light intensity and distance. We also investigated the influence of organic contaminant that resulted in lower activity as well as the effect of a water jet and a high-pressure device on the antibacterial activity. We found that the latter completely removed the coating from the surface, which significantly (*p* < 0.0001) reduced its antibacterial potential. As a conclusion, light intensity and distance does not reduce the efficacy of the polymer, but the presence of organic contaminants does.

## 1. Introduction

Numerous research studies have investigated the potential of photocatalytic materials [1,2] for the purpose of creating self-cleaning surfaces [3], such as textiles [4] or antibacterial surfaces [5,6,7]. Other opportunities for application are to make implants safer [8], treat sewer [9], and reduce the odor caused by gas emission of livestock [10]. As a photocatalyst, TiO_2_ needs UV-light to activate, however, it would be highly beneficial if activation could occur in the visible wavelength, requiring modification of the material’s electrochemical properties. Toward this end, Asahi et al. found TiO_2_ doped with nitrogen to be the most effective, as it changed the bandwidth of the activation wavelength of TiO_2_ such that activation occurred at 500 nm, the range of visible light wavelength [11]. Tahmasebizad et al. described activation of TiO_2_ doped with copper and nitrogen shifting to the visible light range [12]. Tallósy et al., in 2014, found that nano silver similarly impacted the activation of TiO_2_ [13], and in a study from 2015, supported the importance of semiconductor nanoparticles [14]. Tobaldi et al., during their investigations, described the activation of nanostructure TiO_2_ and silver particles photochromism to visible light [15]. Magalhães et al. established that doping with graphene shifted the activating wavelength range [16]. Finally, in the case of TiO_2_ doped with copper in aerosol form, Alotaibi et al. described higher photocatalytic activity [17]. On the other hand, preserving nanomaterial stability in various surfaces is a prominent key factor. To this end, special neutral coatings might be used, such as silica, which not only prevents damage of the treated surface, but also blocks separation of the material [1]. In a study of the relevance of disperse systems, so called nanocomposites enhanced the stability as detailed by Faure et al. [18]. Obregón et al. applied photocatalysts with an electrophoretic procedure to the monitored surface and noticed that photoreactive properties of the coat highly increased, furthermore it had major activity as a self-cleaning surface [19].

TiO_2_ is one of the most widely used photocatalysts [2,20,21]. It is an inexpensive [1] semiconductor material, which has pronounced bactericidal effect [22]. It can be combined with certain semiconductors or materials to be activated at visible light wavelengths [2]. TiO_2_ is often combined with ZnO, which also has photocatalytic potential [23], and may exert its effect in darkness, but this mechanism has not yet been explained [24].

In addition to TiO_2_, ZnO photocatalysts are some of the most attractive semiconductor materials because of their high photosensitivity, high redox potential, low toxicity, high photocatalytic activity, and low cost [25]. However, one of the main disadvantages associated with photocatalysis with ZnO is related to the photo-corrosion that occurs with the irradiation of light over extended periods [26]. TiO_2_ is one of the most resistant substances against corrosion, rendering it a very promising compound to make photoelectrodes and photocatalysts [27]. Fujishima and Honda first described splitting water by using TiO_2_ electrodes, which directed interest to photocatalysis [28]. During the photocatalysis, photons from the light excite TiO_2_, therefore electron migration occurs from the valence band to the conduction band, while electron holes originate [27,29]. Electrons can reduce oxygen, free superoxide anion (-O_2_^-^) and hydrogen peroxide (H_2_O_2_), which are oxygen derivatives capable of oxidizing organic materials. The positively charged holes form hydroxyl radicals (•OH) by reacting with water [29]. Width and energy of band gap determine the conduction properties of compounds. Band gap is between the valence band and conduction band, which in the case of semiconductor TiO_2_, is wide. For this, migration of electrons from the valence band to the conduction band is more difficult [30]. As a consequence, TiO_2_ reacts only with light from the UV (λ < ~400 nm) range, which is small part (~5%) of sunlight. Hence, widening the range of the activation light spectrum is highlighted. Rehman et al. formulated various modification procedures in their study, such as doping with metals or non-metals, or modified surfaces by sensitized polymers [31]. During the doping, band structure is modified by contaminates of different compounds, changing conduction features as follows. Zaleska mentioned various forms of doping, included the opportunity to use metal and non-metal materials [32]. Magalhães et al. investigated a TiO_2_/graphene composite as a form of doping. They attached TiO_2_ nanoparticles to micro-sized graphene plates, which is safer to the body, because it does not penetrate. In addition, graphene is an excellent mobiliser of charge carriers, accelerating the photocatalysis process [16].

The important function of TiO_2_ is that ROS compounds originate as a consequence of its photocatalytic activity, and these reactive species, with high oxidation potential, destroy pathogenic agents by damaging their cell membranes [13,33,34]. Released electrons during photocatalysis react with oxygen which is adsorbed into the surface of TiO_2_. Superoxo (O_2_^−^), peroxide radicals (O_2_^2−)^, and superoxide anion (O_2_^−^) arise, which are made by the dissociation of molecular oxygen as a result of this procedure [35]. However, in addition to these reactive species, hydroxyl radicals (•OH) and H_2_O_2_ are also generated [36]. Different forms of TiO_2_, rutile and anatase have diverse photocatalytic activity [37]. In the case of anatase, release of •OH is higher; on the other hand, rutile has higher ability to generate H_2_O_2_ [38].

Wu et al. investigated point for point the inactivation of *E. coli* by photocatalysis. They identified that bacterial cell membranes are damaged by the process of photocatalysis and found serious lesions in the morphology and inner structure of cells caused by oxidative stress of •OH [39]. Carré et al. had similar results in their studies of elimination of *E. coli* by photocatalysis, and emphasized the photocatalytic effect on lipids and proteins [40]. According to Siddiqi et al., the antimicrobial effect of ZnO, another photocatalyst, is explained by photo excited ZnO nanoparticles permeating across the cell wall and reacting with protein and carbohydrate molecules in the cytoplasm to inactivate them. In addition, release of ROS molecules also contributes to inactivation [41]. In favor of profoundly understanding the photocatalytic bactericide activity of TiO_2_, Takao et al. focused on the peptidoglycan layer of bacterial cell wall during their study. In the course of this investigation, results showed that the presence of peptidoglycan layer increases the bactericide effect of photocatalysis [42]. Rodríguez-González et al., in their review, also described lesions in the bacterial cell wall caused by ROS molecules and metal particles [34].

The aim of our studies was to demonstrate the efficacy of TiO_2_ as a potential alternative to fight antimicrobial resistance, which has significant bacterial reduction capacity against environmental pathogens.

## 2. Materials and Methods

The aqueous photoreactive coating material used in this study consisted of TiO_2_ (Degussa P25) and ZnO (Molar Chemicals) photocatalysts, while the surface immobilization of the particles was ensured by poly(ethyl acrylate-co-methyl methacrylate) (p(EA-co-MMA)) polymer [13].

Photocatalyst paint was applied to the surface with a brush in the afternoon of the day before the test days. The reference bacterial strain used in the tests was ATCC *E. coli* (LGC ATCC-25922, FDA strain Seattle 1946 (DSM 1103, NCIB 12210), Teddington, UK). We chose this bacterium because several studies have shown a link between contamination of food, especially poultry, with *E. coli* strains and human infection [43,44,45,46,47]. Furthermore, this bacteria is commonly used—not because of the disease it can cause—but because it is much easier to cultivate and handle in laboratories because of its low biological safety level (Class I). In most cases, the mediating role of poultry is associated with a higher risk than that of pork or beef [48,49,50]. *E. coli* strains that cause diarrhea of porcine origin have been shown to be able to transmit genes encoding Shiga toxin production with human Stx2 phage [51], and cattle are a major reservoir of Shiga toxin-producing *E. coli* strains [52]. To prepare the *E. coli* stock solution, *E. coli* stored at −80 °C was added to 400 mL of tryptone soy broth (TSB) and incubated at 37 °C for 18–24 h in a thermostat. A 150 mL suspension/surface/test volume of the stock solution was applied to both photocatalyst composite treated and untreated surfaces.

Preliminary tests have shown that it is worth sampling the wall surface for at least 480 min (8 h), during which time the control bacterial count is not significantly decreased. Sampling times were as follows: 00 min before *E. coli* application, 0 min after *E. coli* application, then 15, 30, 60, 90, 120, 240, 360, and 480 min. Sampling was performed with a COPAN (Copan Italia Spa, Brescia, Italy) sterile sampling stick. Samples were grafted onto Petri dishes containing tryptone-soy agar (90 mm × 14.2 mm, beam sterilized, without cams, Biolab Zrt., Budapest). A total of 2400 samples were taken over a period of one month using 36 different combinations of light intensity and spacing. Samples were placed in a 37 °C thermostat and Colony Forming Unit (CFU) counts were performed after 18–24 h incubation time. Samples with >1000 CFU were counted as 1000 CFU for graphical representation, as were samples with >300 CFU; samples with <300 CFU were counted exactly. The standard deviations were calculated in Excel using the following formula: $\sqrt{\frac{\sum^{\;}{\left(x-\bar{x}\right)}^{2}}{\left(n-1\right)}}$; where *x* is the mean value of the average (number1, number2, ...) and *n* is the sample size.

A single light source was placed in the room, and then the designated surface was illuminated from the specified distance for 8 h. For the test, six different wattages of light sources with the same emission spectra (see later) were used: 4 W (320 lux/m^2^), 7 W (600 lux/m^2^), 9 W (806 lux/m^2^), 12 W (1521 lux/m^2^), 18 W (2000 lux/m^2^) and 36 W (3960 lux/m^2^). The wattages of the light sources were converted into lumen per m^2^ (lux/m^2^), the unit of luminous flux. The conversion was performed using the formula Φ V (lm) = P (W) × η (lm/W), where P is the wattage and luminous efficacy (η) is taken to be 90 lm/W.

The choice of distances was based on the fact that light sources in stables are on average 2.5–3 m above the animal contact surfaces. On this basis, we selected six different distances, the smallest being 35 cm, followed by 100 cm, 150 cm, 200 cm, 250 cm and 300 cm.

The light intensity (expressed as W/m^2^) on the irradiated surface of the photoreactive nanohybrid films was measured with a power meter (Thorlabs GmbH, Bergkirchen, Germany). During the measurement, the distance of the light sources from the surface was systematically increased and the corresponding light intensity values were measured. Thus, we determined how light intensity changes with increasing distance from the light source.

When examining the effect of organic pollutants, a layer of *E. coli* suspension from the stock solution was applied. After the bacterial suspension dried, a medium thick layer of pig feces was applied. This process was carried out with care to ensure uniform layer thickness. The parameters most similar to industrial conditions were the primary criteria for selection of the settings, and therefore the two lowest wattage lamps (4 W and 7 W) were selected at the highest distance (300 cm).

The purpose of the mechanical impact test was to check the adherence of the dispersion paint on the wall surface after cleaning impacts. The coating was applied to the designated area on the wall in the afternoon of the day before the test to provide sufficient drying time. In one of the tests, a water jet wash was applied the following morning. The hose was connected to a tap in the room and the surface was washed using normal tap water pressure, no other cleaning equipment or chemicals were used. As a second part of the mechanical impact test, another prepared wall surface was washed with a high-pressure cleaning device on the wall part treated with the designated paint. After washing, and after the wall surfaces had dried, *E. coli* suspension was applied to the surfaces.

In both cases, tap water was used for washing. Washing by both methods was carried out at the same distance from the surface. The mechanical contact time was approximately 15–20 s for both the water jet and high-pressure cleaning. The effect of water jet washing was tested with a 7 W LED lamp at a distance of 300 cm and the effect of high-pressure washing was tested with a 4 W lamp at a distance of 300 cm.

The results were evaluated using the random forest method due to the large number of variables. Random forest is a machine learning method that generates many different decision trees. For numerical variables, the results are averaged, for categorical variables the algorithm counts which result is the most frequent. The decision trees are always generated using only a random subset of variables, so averaging the results of the different decision trees gives a more accurate result than generating a decision tree using all the variables. Information about the importance of the variables is also obtained, which is calculated by the algorithm by looking at the extent to which the hit rates of the decision trees decrease when the variable is omitted. The more the hit rate decreases with the omission of a variable, the more important that variable is.

The hit rate indicates the proportion of data that the model has classified into the correct category. The *p*-values for the variables were determined using the CART (Classification and Regression Tree) method (random forest), in which the algorithm generates a decision tree based on the variables. The analysis was performed using the R program version 4.0.5 (R Foundation for Statistical Computing, Vienna, Austria), and the packages ggplot2, partykit and strucchange.

## 3. Results

### 3.1. Results of the Luminance and Distance Test

The detailed chemical characterization of the photocatalyst particles containing the composite layer used in this study was presented in our previous papers. Briefly, we reported several times that this transparent polyacrylate with good film-forming properties is highly applicable for the surface immobilization of semiconductor photocatalyst particles. At ideal composition (i.e., at optimal photocatalyst/polymer ratio) the polymer component provides appropriate mechanical stability for the composite layer [13,53] while the free photocatalyst particle surfaces are able to exert the photocatalytic properties [54]. Moreover, it was also presented that at higher photocatalyst loading the polymer coverage is not complete and accessible photocatalyst particles can be found on the surface of composite layers and the bacteria obviously preferred the high energy photocatalyst surfaces of the composite layer instead of the low energy polymer [55].

The emission spectra of the light sources used during the experiments can be seen on Appendix A. Each LED lamp has a similar spectrum and emits only in the visible light range. However, Appendix A shows that the measured light intensity values were systematically changed with the increased wattage and it was inversely proportional to the square of the distance from the source. At lowest distance (35 cm) the light intensity of the 4 W LED was only 14.8 W/m^2^, while the intensity of the strongest LED lamp (36 W) was much higher (106 W/m^2^). At larger distances (250–300 cm), however, very low intensity values were measured (<0.5 W/m^2^) in each case.

In the first setting, the 4 W light source test, the photocatalyst composite treated section showed a significant decrease in CFU count on average from the 15th min onwards, until reaching only 2.3% of the baseline count after 30 min. After 90 min, at least 99.5% of the bacteria was eliminated by the photoreactive polymer. On the non-treated surface, there was a confluent increase in CFU counts (Figure 1).

As shown in the statistical analysis, power wattage did not affect overall efficacy. After 15 min, the 36W LED lamp reduced the amount of *E. coli* by the greatest amount at a distance of 250 cm. resulting in 99.7% reduction after 60 min.

### 3.2. Results of Organic Pollutant Effect

Before the application of organic contamination and *E. coli*, no colonies were detected on the photocatalyst-treated surface, whereas 127 (4 W) and 1000 (7 W) CFU were counted on the control surface. Samples taken immediately after application of bacterial suspension and pig manure all had >1000 CFU. All samples taken from non-treated surfaces retained >1000 CFU counts.

Samples taken from the dye-treated area with a 4 W LED lamp after 15 min did not show a significant CFU reduction, however, by 30 min the composite layer showed 78.8% efficiency. From that point on, the amount of CFUs that could be counted steadily decreased, from 89 at minute 60, to 45 at minute 90, and 40 at minute 480. Thus, the photocatalyst layer coated with pig manure illuminated from 300 cm with a 4 W lamp killed 96% of the bacteria overall. When using the 7 W lamp, 205 CFU were counted at 30 min, so 79.5% of the bacteria was killed. At 120 min 112 CFUs were recorded and at 480 min 54 CFUs. Thus, at a distance of 300 cm, when using the 7 W LED lamp, the coating showed an overall efficiency of 94.6% (Figure 2).

### 3.3. Results of the Mechanical Impact Test

When testing the water-washed surface prior to *E. coli* smearing, 300 CFU was counted for the control samples and no colonies were grown for the treated samples. After the bacterial suspension was smeared, >1000 CFU was found on samples taken from both parts. This number did not change at subsequent time points in the control area, and >1000 CFU was still observed at the 480th minute. For the dye-treated area, only 300 CFU was counted at minute 15, which further decreased to 1 CFU at minute 30. This value was maintained until the 120th minute, but no CFUs were found on the sample taken at the 240th minute. From this point on, at minutes 240, 360 and 480, there were also no CFUs, i.e., *E. coli* was completely eliminated (100%) from the surface tested.

After washing with the high-pressure equipment, low CFU values were obtained before the bacterial suspension was applied, 1 CFU for the control and 2 CFU for the coating treated one. After application of *E. coli*, >1000 CFU was observed on all samples. On the untreated control surface, this value was maintained up to 480 min. However, there was no decrease in CFU in the samples taken from the treated surface, with >1000 CFU in the samples taken at the 480th minute (Figure 3).

### 3.4. Statistical Analysis

Our aim was to determine the effectiveness of the photoreactive coating treated surface in reducing bacterial counts. For the treated and untreated wall surfaces tested in parallel, 3 × 3 samples were taken before (00) and 0, 15, 30, 60, 90, 120, 240, 360 and 480 min after the coating was applied. During the experiment, the wall was illuminated at different distances (35 cm, 100 cm, 150 cm, 200 cm, 250 cm and 300 cm) with different wattages of light sources (4 W, 7 W, 9 W, 12 W, 18 W, 37 W). For the first branching of the random forest method, the *p*-value of all variables was less than 0.001 for luminous intensity and distance, but overall, the effects of watt, distance and time were more moderate on the model hit rate compared to the treatment (Figure 4).

When examining the importance of the variables (treatment, distance, wattage, and time), treatment was found to have the highest value (3.3416). The other variables had significantly lower values: watt (0.5265), distance (0.2328), and finally time (0.1638). This also confirms the effectiveness of the treatment, as no effect will be observed if the treatment is abandoned.

In the statistical analysis, we compared the results obtained after the organic pollutant was applied with the results of the luminosity-distance experiment obtained in our first study. We analyzed the values using the random forest method described earlier, where we used only the experimental results of our first study at 4 W and 7 W at a distance of 300 cm. The resulting CFU values were divided into groups for easier statistical analysis and then examined to see what factors influence which category the given bacterial count falls into. The group numbers we chose always indicate the lowest value (e.g., 50 indicates values of 50 or more but less than 100). Except for treatment (*p* < 0.0001), the effect of the variables in the experiment was negligible. Regarding the importance of the variables (treatment, organic pollutant, time, and wattage), treatment was again the most important (6.6135), with negligible effects of organic pollutant (0.7262), time (0.4894), and wattage (0.2927). Again, if the treatment is omitted, no effective reduction in bacterial counts can be expected.

In the third experiment, we first compared high-pressure and water-jet washing, and then compared the results after mechanical treatment with the results without mechanical treatment. The difference between water jet and high-pressure cleaning is significant (*p* < 0.0001), as is the difference in watts (*p* < 0.0001), with the effect of time being negligible. Compared to the control, the results show that high-pressure washing has a significant effect on bacterial counts (*p* < 0.0001), while the effect of normal water jet is negligible (*p* = 0.9998). Among the variables, treatment is also the most important (5.1756), but the effect of the mechanical action itself is significant (3.2573), while the effects of wattage (0.6328) and time (0.1387) are negligible.

## 4. Discussion

The photocatalytic activation of the polymer based composite layer containing TiO_2_ and ZnO particles tested proved to be effective—as also described by Tallósy and Tobaldi [13,15]—our results showed a significant difference between control and treated surfaces (*p* < 0.001), with treatment being the most important statistical variable in each study.

The photocatalytic material was mixed with a polymeric binder, but it was not blocked by the binder, in contrast to the studies of Leng et al. [56]. The efficacy of the binder-free compound was also demonstrated under hospital conditions [57].

It was ascertained that in laboratory circumstances TiO_2_ possessed a bactericide effect. After 30 min treatment in crude water and UV overshine, *E. coli* and *S. aureus* were inactivated at 92.6% and 94.2%. Photocatalytic sterilization potential was effective after 24 h of the beginning of lighting [58]. Nakano et al. investigated multi resistant *Pseudomonas aeruginosa*-t (*P. aeruginosa*) in addition to *E. coli* and *S. aureus* after TiO_2_ treatment, and noticed that Gram-positive bacteria was inactivated at high efficiency, but Gram-negatives required more [59]. Ibrahim et al. demonstrated bactericide activity of TiO_2_ at a rate of 97% against *E.coli* and 95% against *S. aureus* [60]. Previously, Mohl et al. revealed that TiO_2_ has excellent antimicrobial activity against methicillin-susceptible *S. aureus* (MSSA) and methicillin-resistant *S. aureus* (MRSA) [61]. Shimizu et al. detected effective deactivation of *E.coli* with use of composites by carbon nanotubes and TiO_2_ [62]. Janus et al. described that adding TiO_2_ in concrete at 10% by weight has significant antibacterial effect against *E. coli*, it was even able to completely destroy the bacteria [63]. For 5% TiO_2_: Cu multifunctional thin films, Alotaibi et al. formulated that it was successfully able to eliminate antibacterial activity contra *E. coli* and *S. aureus*. Furthermore, researchers showed that photocatalytic efficiency increased greatly because of interstitial and substitutional Cu [17]. Liao et al. described in their review that Cu_2_O/TiO_2_ inactivated 100% of *E. coli* after 60 min [64]. Usage of TiO_2_ coating as an opportunity is confirmed by further research [65,66]. In a *Listeria monocytogenes* biofilm, the number of bacteria decreased 3 log/CFU after 90 min treatment with TiO_2_ and UV-A light [67]. Bartomeuf et al. examined a photoactive layer, which showed increased efficiency to 400 nm threshold against *L. monocytogenes*. In addition, the number of bacteria decreased significantly after 20 min UV-A lighting [68].

Our research has shown that the combination of light source wattages (4 W, 7 W, 9 W, 12 W, 18 W and 36 W) we tested, placed at different distances (35 cm, 100 cm, 150 cm, 200 cm, 250 cm, 300 cm), did not show a significant difference in the efficiency of the photocatalyst layer (*p* < 0.001). This can be explained by the bacteria’s defense mechanisms against ROS compounds circulating in the eye during photocatalysis [69]. No other comparative study of this kind has been published. In their studies, Rizzo et al. described the ability of TiO_2_-containing polymer to completely eliminate *E. coli* bacteria in 10 min for a 250 W light source [70], a result that suggests a much better efficiency in time compared to our studies, but we did not perform 10 min sampling. The difference was probably caused by the different TiO_2_ compound and concentration, and the efficiency was not tested on a surface but in liquid (wastewater). This suggests that it would be worthwhile to investigate the effect of higher wattage light sources and to sample at shorter time intervals for CFU calculation.

The antibacterial effect of ZnO nanoparticles was also investigated against *E. coli*, with use of dispersion-containing and not containing ZnO particles. Dispersion which contained nanoparticles represented powerful antibacterial activity proportional to the concentration of ZnO, and bacteriostatic effect was noticed in 12 mmol/L concentration [71]. Later, the antibacterial effect of ZnO was confirmed by Sirelkhatim et al. [72], and Lallo da Silva et al., and described more powerful photocatalytic activity and biocompatibility compare to TiO_2_ [73]. The antibacterial effect of a nanocomposite made by TiO_2_ and ZnO particles was investigated by Chakra et al. against *E. coli* and *S. aureus* bacteria. Apart from the pronounced antibacterial action, results revealed that TiO_2_ and ZnO particles worked in synergism [74]. Happy et al. made spherical shaped, 60–80 nm sized ZnO nanoparticles with green synthesis and investigated its antibacterial activity. The ZnO compound showed dose-dependent antibacterial effect against *E. coli* (IC_50_ value was 20 μg/mL) [75]. Naseer et al. investigated the antibacterial potential of ZnO nanoparticles made by plant based green synthesis against *E. coli* and *S. aureus* bacteria. In the case of *S. aureus*, standard antibiotics achieved inhibition zone with a range of 4–13 mm, while the ZnO nano compound range was 14–37 mm [76]. The TiO_2_ nanoparticles synthesized by eco-friendly plant extract impacted anti-biofilm effect against *Staphylococcus epidermidis* and *P. aeruginosa* bacteria [77]. Important to note is that excessive usage of nanoparticles increases output to the environment, which may be harmful to the ecosystem. In swine gut whose feed contained ZnO, Ciesinksi et al. found multi resistant *E. coli* bacteria 28.9–30.2% compared to 5.8–14% in the control group. Unfortunately, this refers to the resistant strains [78].

In terms of efficacy against *E. coli*, we found that a significant reduction in CFU counts was observed from the 15th minute onwards, with an average reduction of 97% in CFU counts at 30 min, 98% at 60 min, 99% at 120 min and 99.9% at 480 min compared to the initial CFU counts. Baek et al., however, when testing ZnO, found a reduction of only 72% after 6 h of illumination [79], while Wu et al. in TiO_2_ tests described bacterial count reductions of 20% after 5 min, 90% after 30 min and 100% after 90 min—but TiO_2_ was modified with palladium [39]. Chen et al. found eradication of 92.6% in 30 min in unrefined water and 100% in the same time in isosmotic water [58]. However, Ibrahim et al. observed only a 16% reduction in bacterial counts after 30 min, 67% after 90 min, 80% after 120 min and 97% after 180 min [60]. In comparison, we observed a faster and more effective antibacterial effect, with an average of 97% in 30 min. Janus et al. experienced 100% eradication in as little as 60 min [63], and Liao et al. experienced the same [64]. In contrast, Dong et al. described 100% eradication after 24 h of treatment [80]. Nakano et al., using TiO_2_ on a glass slide, described a reduction of the initial bacterial count of 1.5 × 10^5^ CFU/mL to 1.5 × 10^4^ after 2 h, 1.5 × 10^3^ after 4 h and 1 × 10^2^ after 8 h [59]. In contrast, Alotaibi et al. found a decrease in bacterial counts from an initial 1 × 10^8^ CFU/mL to 1.5 × 10^5^ in 2 h and 1 × 10^3^ in 4 h for copper doped TiO_2_ [17]. However, it should be pointed out that in our studies different concentrations and brightness were used. Based on the published results, it can be concluded that light intensity and distance indeed do not have a significant effect on the photocatalytic antibacterial efficacy of TiO_2_. Our results and those of previous studies are compared in Table 1.

In our organic pollutant study, pig manure reduced the efficiency, but this was not significant (*p* < 0.001). Similar experiences were reported by Chen et al. who investigated the inactivation of *E. coli* by TiO_2_ in unrefined and isosmotic water [58], however, they did not investigate surface contamination. However, Rizzo et al. conducted their study with wastewater, in which *E. coli* bacteria was fully eradicated in 10 min under 250 W illumination and in only 60 min under sunlight illumination [70]. However, no previous study has been described to investigate the effect of contamination of a surface with photocatalyst containing organic fertilizer. It can be concluded that the photocatalyst containing polymer is able to retain its antibacterial activity, but more time is needed for the effect to develop.

The results of the mechanical penetration experiments show that the water jet has little effect, while the high-pressure device completely removed the layer from the wall. Regarding the stability of the photocatalytic compound, Mori et al. described that it is advisable to fix the photocatalyst to a substrate so that it is not damaged by the reactions that occur [1]. It follows from our studies that the treated surfaces can be cleaned with a mild application, but that the strong applications, also used in disinfection, completely remove the photocatalyst layer, and therefore its reapplication is necessary.

## 5. Conclusions

Overall, it can be concluded that the antibacterial activity of the photocatalyst does not depend on either distance or light intensity. When contaminated with organic matter, the effectiveness is slightly reduced, but the difference is only in the length of time required. Washing with water jets is resistant to the coating, but high-pressure treatments remove it. Bacteria can develop several defense mechanisms against light-induced ROS compounds, which may explain the lack of difference in efficacy between different light intensities. Other bacteria of public health importance (*Staphylococcus aureus*, *Pseudomonas aeruginosa*) and fungi (*Aspergillus* spp., *Candida* spp.) should be tested in the future.

## Figures and Tables

**Figure 1 biomedicines-10-02312-f001:**
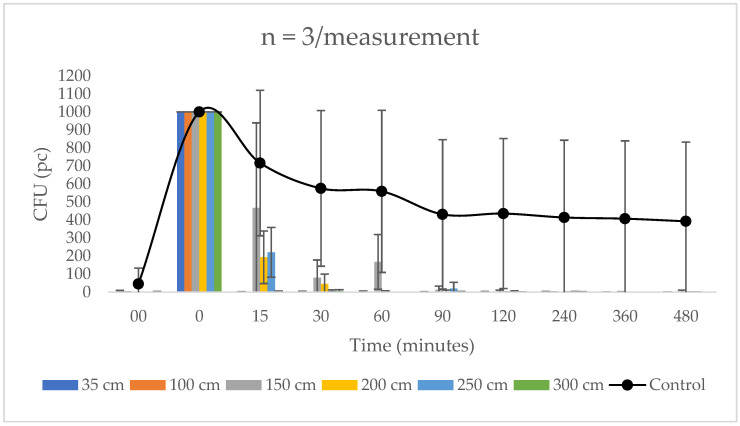
The efficiency of a 4 W light source at each treatment distance compared to the untreated surface. The rate of decrease in the CFU count of a photocatalyst coating-treated surface as a function of time for a 4 W LED lamp at different distances compared to the average CFU count of untreated (Control) surfaces. Starting from 90 min, the bacterial count reduction is 99.5%.

**Figure 2 biomedicines-10-02312-f002:**
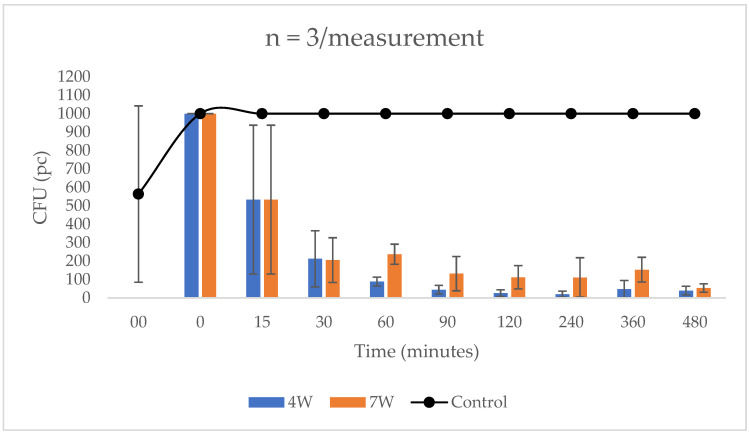
Effect of organic contaminant on photocatalyst layer treated and untreated surface at a distance of 300 cm under illumination with 4 W and 7 W lamps. For samples taken after 15 min, the CFU reduction was not yet significant (78.8% after 30 min for 4 W and 79.5% for 7 W).

**Figure 3 biomedicines-10-02312-f003:**
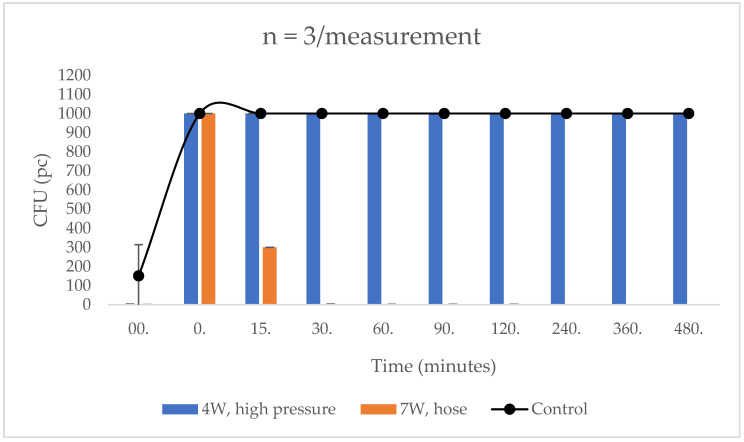
Washing test on the treated and untreated surfaces from a distance of 300 cm. In the case of normal water jet washing, the paint-treated area had only 300 CFU at minute 15, which further decreased to 1 CFU by minute 30, and by minute 120, the bacterial count reduction was 100%. After washing with high pressure equipment, there was no CFU reduction in samples taken from the treated surface even after 480 min.

**Figure 4 biomedicines-10-02312-f004:**
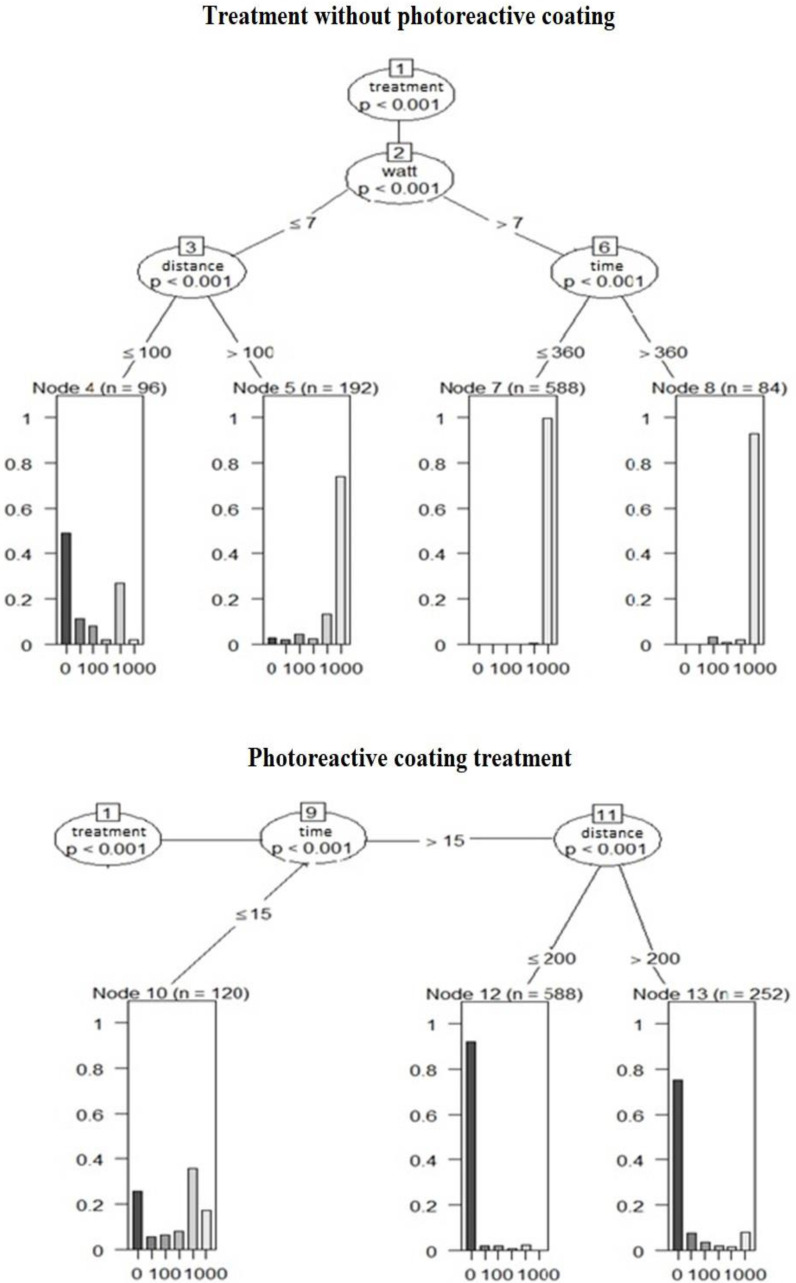
Comparison of control and treated surfaces by random forest method as a function of light intensity and illumination distance. When examining luminance and distance, the *p*-value of all variables was less than 0.001 at the first branching, but overall, the effects of wattage, distance, and time were more moderate on the model hit rate compared to the treatment.

**Table 1 biomedicines-10-02312-t001:** Comparison of the efficacy of TiO_2_-containing polymer composites in *E. coli* with our study and previously published results.

Minutes	5	10	30	60	90	120	240	360	480
Our study	-	-	97%	98%	99.4%	99.9%	99.9%	99.9%	99.9%
Others	20% [39]	100% [70]	92.6% [58]	100% [64]	100% [39]	80.0% [60]	99.0% [59]	72% [79]	99.9% [59]
			90.0% [39]			90.0% [59]			
			92.6% [58]	100% [63]	67% [60]	99.9% [17]	99.9% [17]		
			16.0% [60]						

## Data Availability

Individual measured data can be obtained from the corresponding author.

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
