# Peer review of "Photoreactive Coating Material as an Effective and Durable Antimicrobial Composite in Reducing Bacterial Load on Surfaces in Livestock"

_biomedicines, 2022, doi:10.3390/biomedicines10092312_

Round 1

Reviewer 1 Report

The field of antimicrobial photoreactive coatings already has a large literature. The novelty and significance of the present work lie especially in the combination of approaches and the results should be of practical interest. The Introduction is, however, much too long and unfocused. There is no need to repeat what is already well known in the field -- reference to one or two excellent review papers will suffice. For example Fujishima (2008) (Surf.Sci.Rep. 61;515) is extensive, and Ramsden (2015) (Nanotechnol.Perceptions 11;146) more recent. These reviews also cover mechanisms of bacterial inactivation. On the other hand some aspects are missing. The photostability of TiO2 is mentioned but not that of ZnO. And the random forest method used in the statistical analysis is of great interest and should be mentioned. Strangely no references to this method are given in the Methods section. Regarding light intensity, for work on photocatalysis lumens are inappropriate, what should be given is watts per unit area of the photocatalytic surface. In the Discussion we find "Antibacterial effect of TiO₂ was examined by Chen et al. to E. coli and Staphylococcus 275 aureus (S. aureus) bacterium strains. It was ascertained that in laboratory circumstances 276 TiO₂ possessed bactericide effect." This has been known at least since the pioneering work of Matsunaga et al. in 1985. Anyway such sentences belong in the Introduction. As mentioned, reference to one or two comprehensive reviews removes the need for detailed inclusion of original papers.

Three significant matters need attention

1. Authors mixed photocatalytic material with a polymer binder. One would expect that the photocatalysis then acts on the binder and is blocked. Hence Leng (2013) (Int.J.Inject.Control 9;1) found no antimicrobial effect. On the other hand binder-free material was effective in a hospital -- Reid (2018) (ICHE 39;398). Despite binder, the present authors found an effect. This needs more discussion.

2. Bacteria have defence mechanisms against the ROS generated by photocatalysis. Clemente (2019) (J.Photochem.Photobiol.B193;131) discovered that light intensity below a threshold is more antibacterially effective since it evades these mechanisms. The authors appear to be unaware of this work, which could help to explain their seemingly surprising results regarding light intensity.

3. The authors give no information about the spectral characteristics of their light source. Wattage alone is insufficient. There is superficial mention of wavelength effects in the Introduction, but miss the point that surface states confer visible light activity (see the 2015 review mentioned above). The authors need to attempt quantification of the actinic irradiance.

4. The Conclusions are much too brief and need to be extended in the light of the above 3 matters.

Author Response

Reviewer#1

The field of antimicrobial photoreactive coatings already has a large literature. The novelty and significance of the present work lie especially in the combination of approaches and the results should be of practical interest. The Introduction is, however, much too long and unfocused. There is no need to repeat what is already well known in the field -- reference to one or two excellent review papers will suffice. For example Fujishima (2008) (Surf.Sci.Rep. 61;515) is extensive, and Ramsden (2015) (Nanotechnol.Perceptions 11;146) more recent. These reviews also cover mechanisms of bacterial inactivation. On the other hand some aspects are missing. The photostability of TiO2 is mentioned but not that of ZnO.

Beside TiO2, ZnO photocatalysts are some of the most attractive semiconductor materials because of their high photosensitivity, high redox potential, low toxicity, high photocatalytic activity, and low cost (Appl. Catal. B Environ., 248 (2019), pp. 129-146, 10.1016/j.apcatb.2019.02.017). However, one of the main disadvantages associated with photocatalysis with ZnO is related to the photo-corrosion undergone with the irradiation of light over extended periods (Environ. Sci. Technol., 42 (2008), pp. 8064-8069, 10.1021/es801484x). (see page 2, line 15-21 in the manuscript)

And the random forest method used in the statistical analysis is of great interest and should be mentioned. Strangely no references to this method are given in the Methods section. Regarding light intensity, for work on photocatalysis lumens are inappropriate, what should be given is watts per unit area of the photocatalytic surface. In the Discussion we find "Antibacterial effect of TiO₂ was examined by Chen et al. to E. coli and Staphylococcus 275 aureus (S. aureus) bacterium strains. It was ascertained that in laboratory circumstances 276 TiO₂ possessed bactericide effect." This has been known at least since the pioneering work of Matsunaga et al. in 1985. Anyway such sentences belong in the Introduction. As mentioned, reference to one or two comprehensive reviews removes the need for detailed inclusion of original papers.

The CART (Classification and Regression Tree) method is a synonym for the random forest method, I indicated this in the material and methodology. (see page 4, line 186-189 in the manuscript)

In the discussion section I deleted the problematic sentence: Antibacterial effect of TiO₂ was examined by Chen et al. to E. coli and Staphylococcus aureus (S. aureus) bacterium strains. (no see in the manuscript)

Three significant matters need attention

  1. Authors mixed photocatalytic material with a polymer binder. One would expect that the photocatalysis then acts on the binder and is blocked. Hence Leng (2013) (Int.J.Inject.Control 9;1) found no antimicrobial effect. On the other hand binder-free material was effective in a hospital -- Reid (2018) (ICHE 39;398). Despite binder, the present authors found an effect. This needs more discussion.

The detailed chemical characterization of the photocatalyst particles containing composite layer used in this study was presented in our previous papers. We reported several times that this transparent polyacrylate with good film- forming properties is highly applicable for the surface immobilization of semiconductor photocatalyst particles. At ideal composition (i.e. at optimal photocatalyst/ polymer ratio) the polymer component provides appropriate mechanical stability for the composite layer [13, Colloid Polym Sci (2014) 292:207–217 DOI 10.1007/s00396-013-3063-1] while the free photocatalyst particle surfaces are able to exert the photocatalytic properties [, https://www.frontiersin.org/articles/10.3389/fbioe.2021.709462/full]. Moreover, it was also presented that at higher photocatalyst loading the polymer coverage is not complete and accessible photocatalyst particles can be found on the surface of composite layers and the bacteria obviously preferred the high energy photocatalyst surfaces of the composite layer instead of the low energy polymer [https://doi.org/10.1016/j.clay.2022.106587].  (see from page 4, line 44 to page 5, line 5 in the manuscript)

  1. Bacteria have defence mechanisms against the ROS generated by photocatalysis. Clemente (2019) (J.Photochem.Photobiol.B193;131) discovered that light intensity below a threshold is more antibacterially effective since it evades these mechanisms. The authors appear to be unaware of this work, which could help to explain their seemingly surprising results regarding light intensity.

ROS protection was also incorporated. ].  (see from page 10, line 342-343 in the manuscript)

  1. The authors give no information about the spectral characteristics of their light source. Wattage alone is insufficient. There is superficial mention of wavelength effects in the Introduction, but miss the point that surface states confer visible light activity (see the 2015 review mentioned above). The authors need to attempt quantification of the actinic irradiance.

The light intensity (expressed as W/m2) on the irradiated surface of the photoreactive nanohybrid films was measured with a power meter (Thorlabs GmbH, Germany). During the measurement the distance of the light sources from the surface was systematically increased and measured the corresponding light intensity values. Thus, we determined how light intensity changes with increasing distance from the light sources. (see page 4, line 1-5).

The emission spectra of the light sources used during the experiments can be seen on Fig. S1. Each LED lamp have similar spectra and they emit only in the visible light range. However, Fig. S2 shows that the measured light intsnsity values were systematically changed with the increased wattage and it was inversely proportional to the square of the distance from the source. At lowest distance (35) the light intensity of the 4W LED was only 14.8 W/m2, while the intensity of the strongest LED lamp (36W) was much higher (106 W/m2). At larger distances (250-300), however, very low intesity values were measured (<0,5 W/m2) in each cases. (see page 5, line 6-13 in the manuscript)

  1. The Conclusions are much too brief and need to be extended in the light of the above 3 matters.

In the conclusion section, conclusions were drawn in light of the three issues mentioned above. (see page 11, line 417-420 in the manuscript)

Reviewer 2 Report

In their manuscript "Photoreactive coating material as an effective and durable antimicrobial composite in reducing bacterial load on surfaces in livestock" the authors describe an very interestig feature for use of TiO2.

Also TiO2 ist know since decades for itseffect ("oxidation potential"", the application the authors describe, is very interesting. 

However, I have some remarks:

- the quality of the manuscript could be increaded, if the authors also would focus on fungal contamination, which is a real problem..... (aspergillus, candida, etc) as well as seroius bacteria such as pseudomonas, S. aureus, etc.

- the coblsuion shuld be extended, it is at the current stage very short, and the authors have enough data to give more information also for future direction.

Author Response

Reviewer#2

In their manuscript "Photoreactive coating material as an effective and durable antimicrobial composite in reducing bacterial load on surfaces in livestock" the authors describe an very interestig feature for use of TiO2.

Also TiO2 ist know since decades for itseffect ("oxidation potential"", the application the authors describe, is very interesting. 

However, I have some remarks:

- the quality of the manuscript could be increaded, if the authors also would focus on fungal contamination, which is a real problem..... (aspergillus, candida, etc) as well as seroius bacteria such as pseudomonas, S. aureus, etc.

- the coblsuion shuld be extended, it is at the current stage very short, and the authors have enough data to give more information also for future direction.

Thank you very much for the suggestion to test for fungi. Our previous tests with Escherichia coli are the result of several months of work. It would be beyond the scope of this manuscript to include studies on other bacteria and fungi. On the one hand, this would cause problems of length, and on the other hand, due to the complexity of the studies, it would require at least 6-12 months. In the future, we will certainly consider the feasibility of further studies, but they would be considered for a separate publication anyway.

The conclusion has been extended and the future plans include the testing of additional bacteria and fungi: "Other bacteria of public health importance (Staphylococcus aureus, Pseudomonas aeruginosa) and fungi (Aspergillus spp., Candida spp.) should be tested in the future." (see page 11, line 417-420 in the manuscript)

Reviewer 3 Report

Comments to the Author

This work investigated the antibacterial efficacy of a polymer-based composite layer containing TiO2 and ZnO against E. Coli
. The study demonstrated the remarkable performance of this coating (up to 94% reduction) and concluded that light intensity and distance do not reduce the efficiency of this novel polymer coating. I find this manuscript important for future antibacterial research. The topic falls within the scope of the Biomedicines journal. That being said, my enthusiasm for endorsing the publication of the manuscript in its present form is tempered by several issues and concerns that must be adequately addressed by the authors as detailed below.

1.    In the introduction, a lot of reactions are mentioned in the text. I recommend including the reaction mechanism for at least all-important reactions in this study.

2.    Lines 92-105 focus on the antimicrobial effects against gram-negative bacteria. Can you provide more examples of gram-positive bacteria? It is not clear how this antimicrobial efficiency might change with the type of bacteria.

3.    It is important to include at the end of the introduction the goal of this study and what is going to be done. Why is this study important? What is the novelty of this work? (Line 105)

4.    It would also be better to include chemical characterization results (like FTIR, or XPS) in this study to show that the polymer-based composite layer was synthesized successfully.

5.    Further explanation is needed to explain why E.coli was selected for this study. Is it because previous studies used this type of bacteria?

6.    The quality of the figures should be improved. There is no need to include the title because it is already mentioned under the figure. The addition of tick marks would help read the plots.

7.    Page 9 has a lot of results from the previous studies. I recommend arranging them in a table to see a better correlation between this study and previous studies.

8.    There are some mistakes in the manuscript, and English needs to be further improved.

Author Response

Reviewer#3

This work investigated the antibacterial efficacy of a polymer-based composite layer containing TiO2 and ZnO against E. Coli. The study demonstrated the remarkable performance of this coating (up to 94% reduction) and concluded that light intensity and distance do not reduce the efficiency of this novel polymer coating. I find this manuscript important for future antibacterial research. The topic falls within the scope of the Biomedicines journal. That being said, my enthusiasm for endorsing the publication of the manuscript in its present form is tempered by several issues and concerns that must be adequately addressed by the authors as detailed below.

  1. In the introduction, a lot of reactions are mentioned in the text. I recommend including the reaction mechanism for at least all-important reactions in this study.

There are many other mechanisms that could have been mentioned, but due to the limitations of space, these are highlighted here and are considered to be the most important.

  1. Lines 92-105 focus on the antimicrobial effects against gram-negative bacteria. Can you provide more examples of gram-positive bacteria? It is not clear how this antimicrobial efficiency might change with the type of bacteria.

There are of course examples of Gram-positive bacteria, and quite a few examples are mentioned in the Discussion section. On the contrary, Reviewer 1 suggested a significant shortening of the introduction, so shortening and expanding is controversial for me.

  1. It is important to include at the end of the introduction the goal of this study and what is going to be done. Why is this study important? What is the novelty of this work? (Line 105)

added: "The aim of our studies was to demonstrate the efficacy of TiO2 as a potential alternative to fight antimicrobial resistance, which has significant bacterial reduction capacity against environmental pathogens." (see page 3, line 109-111 in the manuscript)

  1. It would also be better to include chemical characterization results (like FTIR, or XPS) in this study to show that the polymer-based composite layer was synthesized successfully.

The detailed chemical characterization of the photocatalyst particles containing composite layer used in this study was presented in our previous papers. We reported several times that this transparent polyacrylate with good film- forming properties is highly applicable for the surface immobilization of semiconductor photocatalyst particles. At ideal composition (i.e. at optimal photocatalyst/ polymer ratio) the polymer component provides appropriate mechanical stability for the composite layer [13, Colloid Polym Sci (2014) 292:207–217 DOI 10.1007/s00396-013-3063-1] while the free photocatalyst particle surfaces are able to exert the photocatalytic properties [, https://www.frontiersin.org/articles/10.3389/fbioe.2021.709462/full]. Moreover, it was also presented that at higher photocatalyst loading the polymer coverage is not complete and accessible photocatalyst particles can be found on the surface of composite layers and the bacteria obviously preferred the high energy photocatalyst surfaces of the composite layer instead of the low energy polymer [https://doi.org/10.1016/j.clay.2022.106587]. (see from page 4, line 44 to page 5, line 5 in the manuscript)

  1. Further explanation is needed to explain why E.coliwas selected for this study. Is it because previous studies used this type of bacteria?

This is the pathogen of greatest public and animal health importance. This explanation has been added. (see page 3, line 119-125 in the manuscript)

  1. The quality of the figures should be improved. There is no need to include the title because it is already mentioned under the figure. The addition of tick marks would help read the plots.

The titles in the figures have been deleted as requested. (see page 5, 6 and 7 in the manuscript)

  1. Page 9 has a lot of results from the previous studies. I recommend arranging them in a table to see a better correlation between this study and previous studies.

In the present study, a tabular comparison with previous studies is difficult to do and does not give a very clear table, as we have tried to sort. We are absolutely in favour of a tabular overview and support transparency, but we do not believe that this is a practical solution in this article.

  1. There are some mistakes in the manuscript, and English needs to be further improved.
    We improved the English grammar of technical terms.

Round 2

Reviewer 2 Report

- the authors improved the manuscript, however I recommend also to follow the suggestions of the reviewers;

esp- on reviewer asked to present the dat sin form of a table; the authors replied, that they do not think this is necessary .  Maybe the autor think so, bit in a good practise in publication, esp. for publication in this journal, I see this point very necessary;

so, the authors should invest this time to include a table.

- also, in statistcal analyisis presenting the graphs, at leats the SD bars should bei presented in the figure and the number of samples for eac point in the grafic. This is also a standard in presenting the results.

- one sentence also to the use of E. coli.: this bacteria is commonly used - not because of  the disease they  can cause - but because it is much easier to cultivate and handle in laboratories because of its low biological safety level  (Class I) . Other pathegenic germs causing severe diseases are more interesting (class II and III) , but require special safety labs. This should not mixed up with the statement, that most publications use E. Coli.

The authors replied to the other point, I agree; however before publication, at least the table and the correct presentation of data in the graphs should be corrected

Reviewer 3 Report

The authors have made all appropriate corrections.
